# Identification of a Gene Signature That Predicts Dependence upon YAP/TAZ-TEAD

**DOI:** 10.3390/cancers16050852

**Published:** 2024-02-20

**Authors:** Ryan Kanai, Emily Norton, Patrick Stern, Richard O. Hynes, John M. Lamar

**Affiliations:** 1Department of Molecular and Cellular Physiology, Albany Medical College, Albany, NY 12208, USA; kanair@amc.edu (R.K.); nortone@amc.edu (E.N.); 2Koch Institute for Integrative Cancer Research, at Massachusetts Institute of Technology, Cambridge, MA 02139, USA; pstern42@gmail.com; 3Department of Biology, Koch Institute for Integrative Cancer Research, and Howard Hughes Medical Institute, at Massachusetts Institute of Technology, Cambridge, MA 02139, USA; rohynes@mit.edu

**Keywords:** YAP, TAZ, WWTR1, TEAD, metastasis, melanoma, Hippo, gene signature

## Abstract

**Simple Summary:**

Most cancer treatments poison cells non-specifically and can therefore also impact other cells in the body, causing adverse side effects. Targeted therapies block specific proteins that are more essential for cancer cells than for normal cells, limiting side effects. However, targeted therapies are only effective against tumors that rely on the targeted protein for growth or survival. Therefore, it is essential to develop tests that can help predict if a patient’s cancer is dependent upon the targeted protein. Drugs that inhibit the TEAD proteins are currently in clinical trials in cancer patients because many cancers require TEAD function. The goal of our study was to identify genes that are regulated by TEADs in cancer cells and determine if the expression levels of these genes can be used to predict if a cancer is dependent upon TEAD proteins. We identified a set of genes regulated by TEADs in cancer cells and found that the levels of these genes could predict if the cancer cells require TEADs for survival and growth.

**Abstract:**

Targeted therapies are effective cancer treatments when accompanied by accurate diagnostic tests that can help identify patients that will respond to those therapies. The YAP/TAZ-TEAD axis is activated and plays a causal role in several cancer types, and TEAD inhibitors are currently in early-phase clinical trials in cancer patients. However, a lack of a reliable way to identify tumors with YAP/TAZ-TEAD activation for most cancer types makes it difficult to determine which tumors will be susceptible to TEAD inhibitors. Here, we used a combination of RNA-seq and bioinformatic analysis of metastatic melanoma cells to develop a YAP/TAZ gene signature. We found that the genes in this signature are TEAD-dependent in several melanoma cell lines, and that their expression strongly correlates with YAP/TAZ activation in human melanomas. Using DepMap dependency data, we found that this YAP/TAZ signature was predictive of melanoma cell dependence upon YAP/TAZ or TEADs. Importantly, this was not limited to melanoma because this signature was also predictive when tested on a panel of over 1000 cancer cell lines representing numerous distinct cancer types. Our results suggest that YAP/TAZ gene signatures like ours may be effective tools to predict tumor cell dependence upon YAP/TAZ-TEAD, and thus potentially provide a means to identify patients likely to benefit from TEAD inhibitors.

## 1. Introduction

Despite decades of research and significant advances in our understanding of the molecular and cellular events that drive cancer development and progression, cancer remains a leading cause of death worldwide. Most cancer deaths result from metastatic disease, which cannot be effectively treated surgically, so requires systemic therapies. In melanoma specifically, cytotoxic chemotherapies are largely ineffective, so most patients are treated with immune therapies and targeted therapies. Immune therapies have proven successful, but not all patients have durable responses, and some patients are contraindicated due to other health issues. Targeted therapies, such as BRAF and MEK inhibitors, can also be effective, but patients typically develop resistance. Therefore, there is an urgent need for new targeted therapies and a way to predict which patients will benefit from them.

Yes-associated protein (YAP) and transcriptional co-activator with PDZ-binding motif (TAZ) are inappropriately active in many cancer types [1,2,3,4], including melanoma [5,6,7,8]. Hundreds of studies collectively demonstrate that increased YAP or TAZ activity can enhance tumor formation and growth, and promote tumor progression and metastasis [1,3,4,9,10]. YAP and TAZ are negatively regulated by the Hippo Pathway, a serine/threonine kinase cascade that, when active, results in the LATS-mediated phosphorylation of multiple serine residues in YAP and TAZ, promoting either cytoplasmic sequestration or proteasomal degradation [11,12,13,14]. In addition, numerous other cellular pathways can influence YAP and TAZ function either by regulating the Hippo Pathway or through Hippo Pathway-independent mechanisms [15,16,17]. As transcriptional co-activators, YAP and TAZ regulate gene expression programs to mediate their effects on cells. However, both lack DNA binding domains, so they must partner with other transcription factors to regulate target gene expression. Although YAP and TAZ can interact with numerous transcription factors, the TEAD family members play critical roles in YAP/TAZ-dependent gene expression [18,19].

Given the established roles for YAP, TAZ, and TEADs in cancer and other diseases, there has been substantial interest in developing compounds to target these proteins. Although YAP and TAZ have proven difficult to target directly, compounds that target the TEADs, and thus prevent YAP/TAZ-TEAD-mediated transcription, have great promise [19,20]. Indeed, numerous compounds that target YAP/TAZ-TEAD have been described (reviewed in [21]) and three TEAD inhibitors (VT3989, IK-930, IAG933) have recently entered early-phase clinical trials in cancer patients (NCT04665206, NCT05228015, and NCT04857372). An antisense oligonucleotide inhibitor of YAP1 that proved effective in pre-clinical models has also entered clinical trials (NCT04659096). However, as is the case for most targeted therapies, the success of these treatments will depend upon whether tumors that are dependent upon YAP/TAZ-TEAD can be distinguished from those that are not. This will be straightforward for cancers driven by mutations known to cause YAP or TAZ activation, such as NF2-mutant mesothelioma [22], GNAQ-mutant uveal melanoma [23], Epithelioid Hemangioendothelioma (EHE) (which is driven by the oncogenic TAZ-CAMTA1 fusion protein [24,25,26]), and other cancers driven by YAP fusions [27]. However, in most of the cancers where YAP and TAZ play causal roles, mutations or alterations in the Hippo Pathway or YAP and TAZ themselves are rare, so biomarkers that can predict sensitivity to YAP/TAZ-TEAD inhibition will be essential. Given the complexities of how YAP and TAZ are regulated, increased protein expression or nuclear localization do not necessarily indicate increased YAP/TAZ-TEAD activity. The expression of YAP/TAZ-TEAD target genes could provide a more direct readout for YAP/TAZ activation, but whether they can predict sensitivity to YAP/TAZ-TEAD inhibition remains unclear. 

Here, we describe the development of a YAP/TAZ gene signature from metastatic human melanoma cells, which is also highly enriched in cell lines dependent upon YAP, TAZ, or TEADs. Despite being developed from melanoma cell lines, we show that the genes in this signature are YAP/TAZ-dependent in other cancer types, and that this signature is predictive of cancer cell dependence upon YAP, TAZ, and TEADs. This work raises the intriguing possibility that this YAP/TAZ-TEAD gene signature, or others like it, could have diagnostic value to help identify cancer patients likely to benefit from therapies that inhibit YAP/TAZ-TEAD function.

## 2. Materials and Methods

*Cell lines, vectors, and cloning:* Human melanoma cell lines (A375, A375-MA2, A2058) were cultured in growth medium (DMEM+ 10% fetal bovine serum, 2 mM L-glutamine) at 37 °C and 5% CO_2_ and maintained at low passage number. Cells were passaged when sub-confluent and all assays were performed on cells at 60–75% confluence to avoid potential confounding effects of high cell density on YAP/TAZ activity. Cell lines were routinely tested for mycoplasma and other bacterial contaminants. A375 and A2058 cells were obtained from ATCC. A375-MA2 cells were derived from A375 in Richard Hynes’ Lab [28]. All vectors used in this work are listed in Appendix A. If vectors were previously described, received as gifts, or purchased from commercial vendors, the source of the vector is listed [29,30,31,32]. New vectors were generated using standard cloning procedures, and the source constructs used for each insert and vector backbone are indicated. 

*Generation of retrovirus and lentivirus:* Retrovirus and lentivirus were packaged as described previously [33]. Briefly, 293FT cells were plated on 6-well plates at roughly 50% confluence in growth media. After 16–24 h, cells were transfected (according to the manufacturer’s protocol) with a transfection mixture containing 1 μg of viral vector, 0.5 μg of packaging vector (gag/pol), 0.5 μg of coat protein (VSVG), 5 μL of X-tremeGENE™ 9 (Sigma-Aldrich, Milwaukee, WI, USA, Cat#6365779001), and 95 μL of Opti-MEM™ (Life Technologies Corporation (Grand Island, NY, USA), Cat#31985062). The transfection mixture was added to the cells for 24 h, after which, the mixture was removed, and the cells were fed with fresh growth media. Culture supernatant was collected and filtered through a 0.45 μm filter 24 h later. For stable transduction, cells at roughly 60–80% confluence were incubated with viral supernatant diluted 1:1 with fresh growth medium and Polybrene (Sigma-Aldrich, Cat#45-H9268) (final concentration 8 µg/mL) for 24 h and then viral supernatants were removed, and cells were fed with fresh growth medium and stably selected with the appropriate antibiotic.

*RNAi:* siRNA experiments used Horizon Discovery SMARTPools^TM^ (Horizon Discovery, Waterbreach, UK) and included a non-targeting control siRNA (Horizon Discovery ON-TARGETplus^TM^ Non-targeting siRNA #1, Cat#D-001810-01-05) or SMARTPools^TM^ targeting human YAP (Horizon Discovery ON-TARGETplus^TM^ Human YAP1 (Entrez gene ID 10413) siRNA, Cat#L-012200-00-0010) and human TAZ (Horizon Discovery ON-TARGETplus^TM^ Human WWTR1 (Entrez gene ID 25937) siRNA, Cat#L-016083-00-0010). Cells were plated at 4 × 10^5^ cells per 6 cm plate and cultured in growth medium for 24 h. A transfection mixture containing 9 μL (90 pmol) of siRNA, 27 μL of Lipofectamine™ RNAiMAX (Life Technologies Corporation Cat#13778075) and 900 μL Opti-MEM^TM^ was setup according to the manufacturer’s protocol. For the combined knockdown of both YAP and TAZ, 4.5 μL (45 pmol) of each siRNA SMARTPool^TM^ was used so that the total volume of siRNA remained at 9 μL (90 pmol). Twenty-four hours after the transfection, the cells were trypsinized and plated for Western blots or qPCR and cultured for an additional 48 h in growth medium before lysing as described below. 

*YAP/TAZ transcriptional activity:* The YAP/TAZ-TEAD transcriptional reporter assays utilized a TEAD reporter construct (pGL3-5xMCAT(SV)-49 [30,33] that consists of 5 repeats of a TEAD binding element upstream of a minimal SV40 promoter that drives expression of the gene that encodes Firefly Luciferase. A control vector encoding a constitutively expressed *Renilla* luciferase (PRL-TK (Promega, Madison, WI, USA, Cat#E2231)) is co-transfected and used for normalization. Cells were plated at 1.5 × 10^5^ cells per 12-well in duplicate in 1 mL of growth media. After 24 h, cells were co-transfected with a transfection mixture containing 800 ng of a 20:1 mixture of pGL3-5xMCAT(SV)-49 and PRL-TK, 2.754 μL/well of Lipofectamine^TM^ 3000, 1.62 μL/well of the P3000 reagent (Life Technologies Corporation, Cat#L3000001), and 100 μL of Opti-MEM^TM^. After 24 h, luciferase activity was assayed using the Dual-Luciferase Reporter Assay System (Promega, Cat#E1910) and normalized luciferase levels were calculated as described previously [33]. For some experiments, cells were transfected with siRNAs or infected with viral constructs prior to assaying luciferase activity. 

*Western blotting and qPCR:* For Western blots, cells were plated at 1.5 × 10^5^ cells per 6-well plate in full growth media. Forty-eight hours later the cells were lysed in Cell Lysis Buffer (Cell Signaling Technology, Danvers, MA, USA, Cat#9803) containing Pierce™ Protease Inhibitor Mini Tablets (Thermo Fisher Scientific, Rockford, IL, USA Cat#88665) and Pierce™ Phosphatase Inhibitor Mini Tablets (Thermo Fisher Scientific, Cat#88667). The protein concentration was determined using the Pierce™ BCA protein assay kit (Thermo Fisher Scientific, Cat#23225) and equal protein (20–30 μg) was subjected to 10% SDS-PAGE, transferred to PVDF membranes, and assayed by Western blot. The following primary antibodies were used at 1:1000 dilutions: total YAP (D8H1X) XP (Cell Signaling Technology, Cat#14074); total YAP (Cell Signaling Technology, Cat#4912,); total TAZ (V386) (Cell Signaling Technology, Cat#4883); and GAPDH Cell Signaling Technology (Cat#2118). The following horseradish–peroxidase-conjugated secondary antibodies were used at 1:5000 dilutions: goat anti-rabbit IgG (Thermo Fisher Scientific, Cat#31460) and goat anti-mouse IgG (Thermo Fisher Scientific, Cat#32430). Primary and secondary antibodies were diluted in 5% BSA. Western blot images were captured with a Fujifilm LAS-3000 gel imager. For qPCR, cells were plated at 1.0 × 10^5^ cells per 6-well plate in full growth medium and cultured for 48 h, then lysed with TRIzol^®^ (Life Technologies Corporation, Cat#15596018) and RNA was isolated following the manufacturer’s protocol. Following the manufacturer’s protocol, cDNA was made from 200 ng of the total RNA using qScript^®^ cDNA SuperMix (QuantaBio, Beverly, MA, USA, Cat#95048). qPCR reactions were carried out on 2 μL of cDNA, using 2 pmol of each primer (Appendix A) and 10 µL of PerfeCTa SYBR Green^®^ Fast Mix (QuantaBio, Cat#101414-270). The reaction mixture was brought to a total volume of 20 µL with nuclease-free water. qPCR reactions were run using a CFX Connect real-time PCR detection system according to the manufacturer’s instructions (Bio-Rad, Hercules, CA, USA, Cat#1855201). PCR conditions were 95 °C for 30 s, followed by 40 cycles of 95 °C for 10 s and 60 °C for 30 s, followed by a melt temperature analysis. For data processing, the Bio-Rad CFX Maestro software was used to calculate the fold change in mRNA for each indicated gene for each sample relative to a pre-determined control sample using the ΔΔCt method and GAPDH as a reference gene.

*In vivo metastasis assays:* The Albany Medical College Institutional Animal Care and Use Committee approved all mouse studies. Mice were housed in specific pathogen-free conditions in the Albany Medical College Animal Resources Facility, which is licensed by both the USDA and the NYS Department of Health, Division of Laboratories and Research, accredited by the AAALAC, and has assurance approval from the Office of Laboratory Animal Welfare (identification number is D16-00062 (A3099-01)). These studies used immunocompromised NOD/Scid mice (NOD/MrkBomTac-Prkdcscid, Taconic, Rensselaer, NY, USA). To assay metastatic colonization, fluorescently-labeled A375 cells expressing the indicated constructs were injected into the lateral tail veins of mice at 1 × 10^6^ cells per mouse in 100 µL of PBS, and after 6 weeks, mice were euthanized, and lung metastases were counted using an Olympus SZX9 fluorescent stereomicroscope (Evident Scientific, Waltham, MA, USA). For this, lung lobes were separated and placed in a tissue culture plate and visible GFP-positive metastases were counted on both sides of each lobe by a researcher blinded to which cells were injected. Representative images were taken using a Lumenera Infinity 3S camera (Teledyne Lumenera, Ottawa, ON, Canada).

*RNA-seq and differential gene expression analysis:* A375 cells stably expressing the indicated constructs were cultured in growth medium for 48 h and then lysed with TRIzol^TM^ (Life Technologies Corporation, Cat#15596018), and RNA was isolated following the manufacturer’s protocol. All cultures were of comparable cell density (60–75% confluence) at the time of lysing. cDNA was prepared with Illumina TruSeq chemistry and libraries were prepared using SPRIworks (Beckman Coulter, Brea, CA, USA) and sequenced using TruSeq SBS Kit v3 on the Illumina HiSeq2000 (Illumina, San Diego, CA, USA). Sequence reads were aligned to the UCSC known genes version 2012 hg19 human assembly using bowtie2 version 2.0.0-beta6 [34] and tophat version 2.0.4 [35]. Transcript assembly, gene summary, and differential expression was performed using cufflinks version 2.0.2 [36]. RNA-seq data are available in the NCBI Gene Expression Omnibus (GSE234083). Differentially expressed genes in the Control siRNA vs. YAP/TAZ siRNA transfected SK-MEL-28 or WM3248 cells were identified from a publicly available gene expression dataset (GSE68599). This dataset was downloaded into Exatlas and genes up- or downregulated in siControl vs. siYAP/siTAZ SK-MEL-28 or WM3248 cells were identified using a fold change >2 and an FDR < 0.05 as the cutoffs for the analysis. Genes differentially expressed in MeWo cells expressing YAP5SA vs. Control and their fold-change were taken directly from the supplemental data in this publication [6]. 

*Bioinformatic Analyses:* Publicly available gene expression datasets downloaded from the NCBI Gene Expression Omnibus (https://www.ncbi.nlm.nih.gov/geo/, accessed on 2 February 2022) are listed in Appendix A, Tab 3. Analyses of datasets downloaded from The Cancer Genome Atlas (TCGA) (https://www.cancer.gov/tcga/, accessed on 6 December 2022), The Broad Institute Cancer Dependency Map (DepMap) Portal (https://depmap.org/portal/, accessed on 5 September 2023), or The ENCODE Project Database (https://www.encodeproject.org/, accessed on 9 December 2022) [37] are described below. Metascape [38] was accessed via https://metascape.org/gp/index.html#/main/step1 (accessed on 19 April 2022). 2. For analysis with GSEA (version 4.3.2), we downloaded gene expression datasets from NCBI-GEO, TCGA, or DepMap, and generated .gct files and .cls phenotype files for the indicated comparisons. GSEA [39,40] was run to test for the enrichment of genesets described in Appendix A, Tab 4 using the appropriate CHIP platform and default settings. Probe/transcript Ids were collapsed to gene symbols, 1000 permutations were run, and the permutation type used was “phenotype” if the number of samples in each group was 8 or more, or “geneset” if the number of samples in each group was less than 8. Rank-ordered lists were generated by running GSEAPreranked (version 4.3.2) with the same settings described above. Heatmaps were generated using MORPHEUS (https://software.broadinstitute.org/morpheus/, accessed on 2 September 2023). To calculate Z-Scored expression, TPM values were converted to log2 (1 + x) and Z-Scored in MORPHEUS. Similarity matrices were generated from the Z-Scored expression values by first calculating the Spearman’s Rank Correlation of each gene with either *CTGF* or *CYR61* using the nearest neighbor function and 1000 permutations. Genes were then sorted by Spearman’s Rank Correlation values (highest to lowest) and the Similarity Matrix function was used to generate the Spearman’s Rank Correlation for each pairwise comparison of genes in the dataset.

*ENCODE:* Conservative IDR Threshold Peaks for the indicated ChIP-Seq datasets were downloaded from the ENCODE Portal [37] (https://www.encodeproject.org/, accessed on 9 December 2022) (Appendix A, Tab 5). The ChIPseeker package in R (version 1.38.0) [41] was then used to annotate peaks to protein coding genes, using the GRCh38 genome assembly, with the transcriptional start site region (tssRegion) parameter set as +/−1 kb and the genomicAnnotationPriority parameter set to the default order: (“Promoter”, “5UTR”, “3UTR”, “Exon”, “Intron”, “Downstream”, “Intergenic”). ENTREZ IDs for annotated peaks were converted to gene symbols using the ENSEMBL Database Homo Sapiens V86 as a reference. 

*DepMap:* The RNA-seq data file “OmicsExpressionProteinCodingGenesTPMLogp1.csv” was downloaded from “DepMap Public 23Q2 Primary Files” on The Broad Institute Cancer Dependency Map (DepMap) Portal (https://depmap.org/portal/, accessed on 5 September 2023). Dependency scores for *TEADs 1–4*, *YAP1*, and *WWTR1* (TAZ) were downloaded from DepMap. The RNA-seq data from the 1019 cell lines that have reported *YAP*1, *WWTR1*, and *TEADs* dependency scores were analyzed for the enrichment of genesets using GSEA analysis, as described above. To generate ROC curves, the GSVA package in R [42] was used to calculate enrichment scores for the indicated genesets in the indicated cell lines, and the resulting GSVA enrichment scores were used to perform ROC curve analyses using MedCalc for Windows, version 22.013 (MedCalc Software, Ostend, Belgium). Cell lines were considered dependent upon a gene if the Chronos Dependency Score was ≤−0.65. 

*The Cancer Genome Atlas (TCGA):* RNA-seq data from 303 human melanoma samples in the TCGA-Skin Cutaneous Melanoma (SKCM) project were downloaded from the NCI Genomic Data Commons Data Portal. Only datasets with an RNA integrity number between 7 and 10 were used. The transcripts per million (TPM) data for each sample were compiled and used for GSEA analysis, to generate heatmaps, and for similarity matrix calculations. 

*Statistical Analyses:* Statistical analyses were performed in GraphPad Prism (version 10.1.1). The statistical tests used to determine significance and the number of ns are indicated in the legends. All scatter plots show mean ± S.D unless noted otherwise in the legend.

## 3. Results

### 3.1. Identification of a YAP/TAZ Gene Signature in Metastatic Melanoma Cells

The LATS-mediated phosphorylation of YAP on serine 127 and serine 397 results in cytoplasmic sequestration and proteasomal degradation, respectively, and the serine-to-alanine mutation of these residues is known to render YAP insensitive to LATS-mediated inhibition [11,14]. Our previous work in melanoma and breast cancer demonstrated that mutant forms of YAP unable to be phosphorylated on one or both of these LATS phosphorylation sites promote tumor growth and metastasis, and that this requires TEAD interaction [31]. This suggests that YAP drives metastasis by enhancing TEAD-dependent gene expression. Here, we show that the expression of mutant forms of YAP that are insensitive to LATS-mediated cytoplasmic sequestration (YAP^S127A^) or to both LATS-mediated cytoplasmic sequestration and proteasomal degradation (YAP^2SA^) in A375 human melanoma cells increased TEAD transcriptional activity and enhanced metastatic colonization of the lung (Figure 1A–D). However, mutant forms of YAP^2SA^ unable to bind to TEADs due to a serine-to-alanine mutation at residue 94 [43] (YAP^2SA,S94A^) or that lack the transactivation domain (YAP^2SA^-∆TA) did not promote TEAD transcriptional activity or metastatic colonization (Figure 1A–D), suggesting that YAP-TEAD promotes metastasis by regulating the transcription of target genes. To elucidate changes in gene expression in cells rendered metastatic by YAP activation, we performed RNA sequencing on A375 cells stably expressing a control vector, YAP^2SA^, YAP^S127A^, or YAP^S127A, S94A^ [31], which cannot bind TEADs. Stable YAP^2SA^ expression upregulated 903 genes and downregulated 783 genes. GSEA analysis with several Hippo Pathway-related gene sets from the Molecular Signature Database [39] revealed the enrichment of genes associated with YAP activation in the A375-YAP^2SA^ cells compared to the control A375 cells (Figure 1E). The YAP^S127A^ mutant only upregulated 176 genes and downregulated 10 genes (Figure 1F and Appendix A). The fact that TEAD transcriptional activity is more dramatically induced by YAP^2SA^ than it is by YAP^S127A^ (Figure 1B) likely explains why substantially more genes were differentially expressed in a statistically significant manner in the YAP^2SA^-expressing cells than in the YAP^S127A^-expressing cells. Indeed, most of the differentially regulated genes in the A375-YAP^2SA^ cells were also upregulated or downregulated in the A375-YAP^S127A^ cells, but to a lesser magnitude (Figure 1G and Appendix A). Interestingly, only eight genes were differentially expressed in the YAP^S127A,S94A^-expressing cells compared to control cells (Figure 1F,G and Appendix A), suggesting that in A375 cells, most gene expression changes induced by YAP are mediated by TEADs.

To test which YAP-dependent genes we identified in A375 cells were also regulated by YAP or TAZ in other metastatic melanoma cell lines, we used publicly available gene expression datasets (Appendix A, Tab 3). This included MeWo cells rendered metastatic by the expression of LATS-insensitive YAP^5SA^ [6] and two metastatic melanoma cell lines (SK-MEL-28 and WM3248) with both YAP and TAZ knocked down [44]. Genes were considered upregulated by YAP/TAZ if they were induced in A375-YAP^2SA^ or MeWo-YAP^5SA^ or if they were downregulated by YAP/TAZ siRNA in SK-MEL-28 or WM3248. Conversely, genes downregulated in A375-YAP^2SA^ or MeWo-YAP^5SA^ or upregulated by YAP/TAZ siRNA in SK-MEL-28 or WM3248 were considered downregulated by YAP/TAZ. Although many of the differentially expressed genes in each cell line were distinct, we identified a 132-gene YAP/TAZ signature consisting of genes that were upregulated (80 genes) or downregulated (52 genes) in A375-YAP^2SA^ cells and at least two of the other metastatic melanoma cell lines (Figure 2A,C and Appendix A). This signature included several known YAP/TAZ target genes (Figure 2B) and was enriched with genes involved in cell migration, cell adhesion, angiogenesis, core matrisome, and cancer pathways (Figure 2D).

A parallel study from our lab found that the highly metastatic human melanoma cell lines A375-MA2 (hence called MA2) and A2058 require YAP and TAZ for the metastatic colonization of the lungs (Warren et al. in preparation). To test if the YAP/TAZ target genes that we identified were also YAP/TAZ-dependent in these metastatic melanoma cell lines, we transfected MA2 and A2058 cells with either a control siRNA or siRNA pools targeting both YAP and TAZ. Western blots confirmed that YAP and TAZ protein levels were dramatically reduced in the siYAP/TAZ-transfected cells (Figure 3A). As expected, this led to a substantial inhibition in TEAD-mediated transcriptional activity (Figure 3B). We used qPCR to analyze the mRNA expression of several of the upregulated genes we identified in Figure 2 and found that YAP/TAZ knockdown reduced the expression of most of the genes in both cell lines (Figure 3C), indicating that these genes are YAP/TAZ-dependent in these metastatic melanoma cell lines.

### 3.2. YAP/TAZ Signature Genes Are TEAD-Dependent

Many YAP/TAZ-dependent processes in cancer cells are mediated by TEADs. Consistently, we found that the majority of the genes that were differentially expressed in A375-YAP^2SA^ cells also changed in a similar manner in A375-YAP^S127A^ cells, but did not change in A375 cells expressing the YAP^S127A,S94A^ mutant that cannot bind TEADs (Figure 1G). This indicates that the YAP-dependent regulation of these genes is TEAD-dependent in A375 cells. We next tested if YAP/TAZ-dependent genes from our signature were also TEAD-dependent in additional melanoma cell lines. We blocked TEAD-mediated transcription in MA2 and A2058 cells by expressing either TEAD1^Y421E^, a dominant negative form of mouse TEAD1 (DN-TEAD1) [45], or a genetic TEAD inhibitor (TEADi) [32], and performed qPCR. As expected, both TEADi and DN-TEAD1 dramatically reduced the activity of the YAP/TAZ-TEAD reporter in both MA2 and A2058 cells (Figure 4A), indicating that these constructs effectively blocked TEAD-mediated transcription. The expression of either TEADi or DN-TEAD1 also reduced the expression of the genes from our YAP/TAZ signature that we tested (Figure 4B), confirming that these genes are dependent upon TEADs in these cells. 

We next used GSEA to analyze available gene expression data from an invasive human melanoma cell line (MM047) in which all four TEADs were knocked down [46] to look for enrichment of our YAP/TAZ signature genes. Many of the 80 upregulated genes (YAP/TAZ Up) were strongly enriched (i.e., upregulated) in the control cells when compared to the TEAD knockdown cells (Figure 4C), indicating that these genes are TEAD-dependent. Surprisingly, only a few of the 52 downregulated genes (YAP/TAZ Down) were enriched in TEAD knockdown cells, suggesting that unlike the upregulated genes, most of the downregulated genes were not TEAD-dependent in MM047 cells (Figure 4C). Next, we analyzed several TEAD ChIP-seq datasets available in the ENCODE Database using the ChIPSeeker package to determine which of our YAP/TAZ signature genes had associated TEAD peaks. Most of the 80 YAP/TAZ Up genes had TEAD peaks in several of the ChIP-seq datasets that we analyzed (Figure 4D), indicating that TEADs were bound to genomic regions associated with these genes. Although most of the YAP/TAZ Down genes also had TEAD peaks, fewer ChIP-seq datasets had peaks for the YAP/TAZ Down genes compared to the YAP/TAZ Up genes (Figure 4D), which could suggest that the binding of TEADs to regulatory regions is more consistent for the genes upregulated by YAP and TAZ than it is for the genes downregulated by YAP and TAZ. Collectively, these results show that many of the upregulated genes in our YAP/TAZ signature are TEAD-dependent in melanoma cell lines. 

### 3.3. Upregulated YAP/TAZ Signature Genes Are Predictive of Cancer Cell Dependency upon YAP/TAZ-TEAD

With TEAD inhibitors currently in clinical trials, developing an accurate way to predict sensitivity to YAP/TAZ-TEAD inhibition is necessary. To test if our YAP/TAZ signature has predictive value in melanoma, we used Chronos Dependency Scores from the DepMap Portal to identify melanoma cell lines that are dependent upon YAP, TAZ (which is encoded by the *WWTR1* gene), or TEADs for viability, and then analyzed the expression of our YAP/TAZ signature genes in these cells. Of the 62 melanoma cell lines in the DepMap Portal, 17 are dependent upon TAZ, 2 are dependent upon YAP, 9 are dependent upon TEAD1, and 1 is dependent upon TEAD4 (Appendix A and Appendix A). None of the cell lines are dependent upon TEADs 2 or 3. We performed GSEA analysis to test for enrichment of our YAP/TAZ Up and YAP/TAZ Down genesets and other published YAP Up genesets [47,48,49] in the cell lines dependent upon either YAP or TAZ (YAP/TAZ) or upon any of the TEADs. The YAP/TAZ Up geneset was highly enriched (i.e., upregulated) in the TEAD-dependent and YAP/TAZ-dependent cell lines (Figure 5A). The Cordenonsi YAP Up and Wang YAP Up genesets also showed enrichment in the dependent cell lines; however, these genesets had lower normalized enrichment scores, higher false discovery rates, and lower % of genes in the leading edge indicating that they were not as strongly enriched as our YAP/TAZ Up geneset (). Our YAP/TAZ Down geneset showed only a slight negative enrichment in TEAD-dependent and YAP/TAZ-dependent cell lines, suggesting that upregulated genes may be better predictors of YAP/TAZ-TEAD dependence than downregulated genes. 

The above data suggest that cell lines dependent upon YAP/TAZ or TEADs have higher expression of the upregulated genes in our YAP/TAZ signature compared to cell lines not dependent upon YAP/TAZ or TEADs. Consistently, although there were a few exceptions, most of the YAP/TAZ Up genes were more highly expressed in the cell lines with the lowest dependency scores (i.e., the most dependent cell lines) (Figure 5B and Appendix A). We next tested how well our YAP/TAZ Up geneset could predict the dependency of these melanoma cell lines upon YAP/TAZ or TEADs relative to the other published YAP Up genesets. Gene Set Variance Analysis (GSVA) was used to score the relative enrichment of each geneset in each melanoma cell line (see Appendix A, Tab 5), and then receiver operating characteristic (ROC) curves were generated to determine how well each geneset could predict dependency upon YAP/TAZ or TEADs. Although all three genesets were predictive of dependency upon TEADs or YAP/TAZ, our YAP/TAZ Up and the Wang YAP Up genesets were the most predictive (higher AUC and Youden Index values) (Figure 5C). Overall, these data show that the enrichment of our entire YAP/TAZ up geneset can help determine if melanoma cell lines are dependent upon YAP/TAZ-TEAD, raising the intriguing possibility that this signature could be used to help identify melanoma patients that may respond to TEAD inhibition.

### 3.4. Upregulated YAP/TAZ Signature Genes Strongly Correlate with YAP/TAZ Activation in Human Melanomas

Our analysis so far has used gene expression data from cell lines cultured in vitro to identify YAP/TAZ-TEAD target genes and test their predictive value. We next sought to determine if our YAP/TAZ signature genes are also YAP/TAZ-dependent in human melanoma, so we analyzed RNA-seq data from The Cancer Genome Atlas (TCGA) Human Skin Cutaneous Melanoma (SKCM) project. Although increases in *YAP1* or *WWTR1* (TAZ) mRNA expression in a tumor could result in increased protein expression and thus elevated YAP/TAZ activity, most of the regulation of YAP and TAZ occurs at the post-translational level, so *YAP1* or *WWTR1* mRNA expression is likely not the best readout for YAP/TAZ activity. Consistently, when we analyzed the TCGA-SKCM dataset, we found a poor correlation between *YAP1* mRNA expression and the mRNA expression of several established YAP/TAZ target genes found in existing YAP Up genesets [47,48] (*CTGF/CCN2*, *CYR61/CCN1*, *ANKRD1*, *CRIM1*, *DDAH1*, and *F3*) (Appendix A). There was also not a significant difference in the expression of most of these genes in tumors with high *YAP1* mRNA expression when compared to tumors with low *YAP1* mRNA expression (Appendix A). *WWTR1* mRNA expression showed a stronger correlation (i.e., higher Pearson Correlation Coefficient values) with YAP/TAZ target genes, and the expression of these genes was generally elevated in tumors with high *WWTR1* mRNA expression when compared to tumors with low expression (Appendix A). However, these genes showed a much stronger correlation with the mRNA expression of the established YAP/TAZ target gene *CTGF*/*CCN2* and their expression was significantly higher in *CTGF* high tumors compared to *CTGF* low tumors (Appendix A). Consistently, genes in published YAP Up genesets [47,48] were more upregulated in *CTGF* high vs. *CTGF* low tumors than they were in the *YAP1* or *WWTR1* high vs. low tumors (Appendix A). Furthermore, DepMap Portal data show that the expression of *YAP*1, *WWTR1*, or *TEADs* did not strongly correlate with dependence upon YAP, TAZ, or TEADs (Appendix A). Thus, the expression of YAP/TAZ target genes such as *CTGF* and *CYR61* appears to be a more reliable indicator of increased YAP/TAZ activity than the mRNA expression of *YAP1* and *WWTR1* themselves. 

Given the above results, we next tested which of our YAP/TAZ signature genes were correlated with the expression of *CTGF* or *CYR61* in the TCGA-SKCM tumors. GSEA analysis revealed that many of the YAP/TAZ Up genes were enriched in tumors with high *CTGF* or *CYR61* mRNA expression compared to tumors with low expression (Figure 6A). Consistently, the expression of most of the YAP/TAZ Up genes was significantly higher in tumors with high *CTGF* or *CYR61* expression compared to tumors with low expression (Figure 6B,D). To determine how strongly the expression of YAP/TAZ Up genes correlated with each other, we analyzed the TCGA-SKCM data using Spearman’s Rank Correlation similarity matrices. This revealed a large subset of the YAP/TAZ Up genes that showed a strong correlation with *CTGF* and *CYR61*, as well as with each other (Figure 6C,E). Of the 80 YAP/TAZ Up genes, 49 were strongly correlated with the mRNA expression of *CTGF*, *CYR61*, or both (Figure 6F,G and Appendix A). In contrast, the YAP/TAZ Down genes did not show a strong negative correlation with *CTGF* or *CYR61* mRNA expression. In fact, some of these genes were positively correlated with *CTGF* or *CYR61* (Appendix A and Appendix A). This suggests that although these genes are downregulated by YAP and TAZ activation in metastatic melanoma cells in vitro, they are not consistently downregulated by YAP or TAZ in human melanomas in vivo. Similarly, the data presented above (and below) demonstrate that, overall, YAP/TAZ Down genes are not as consistently regulated by YAP and TAZ in different cell lines as the YAP/TAZ Up genes are. Nevertheless, our data suggest that most of the YAP/TAZ Up genes in our signature are strongly correlated with YAP/TAZ activity in tumors isolated from melanoma patients. 

### 3.5. YAP/TAZ Signature Genes Are Predictive of Dependence upon YAP/TAZ-TEAD in Other Cancer Types

Although YAP and TAZ promote tumor growth and progression in multiple cancers (reviewed in [1,3,4,9]), it is unknown to what degree YAP/TAZ-dependent gene expression is conserved across different cancer types. Since TEADs recruit YAP or TAZ to consensus motifs in the enhancer and promoter regions of target genes, YAP/TAZ-TEAD-regulated genes in different cancers are likely to overlap to some degree. Indeed, several of the genes in our YAP/TAZ signature were established as YAP/TAZ targets in other cell types. However, tissue-specific transcription factors and epigenetic regulators could also influence which genes YAP and TAZ regulate in each cancer type. To determine which of our YAP/TAZ signature genes are also YAP/TAZ-responsive in other cancers, we analyzed 31 available gene expression datasets generated from cancer cell lines in which YAP and/or TAZ expression or function was altered (Appendix A) [6,44,46,50,51,52,53,54,55,56,57,58,59,60,61,62,63,64,65,66,67,68]. Many of the 132 YAP/TAZ signature genes were also regulated by YAP/TAZ in several of the other cancer cell lines (Figure 7A and Appendix A). The upregulated genes in our YAP/TAZ signature were significantly more conserved than the downregulated genes. Intriguingly, there was not a single gene that was YAP/TAZ-dependent in every cell line, and most of the genes were only YAP/TAZ-responsive in a subset of the cancer cell lines (Figure 7A and Appendix A). This suggests that although most of the YAP/TAZ Up genes in our signature are regulated by YAP or TAZ in other cancers, which subsets of genes respond to changes in YAP/TAZ activity in each cancer cell line varies significantly. 

YAP Up genesetsare often used for geneset enrichment analysis to test for increased YAP/TAZ activity in cells or tissues. To test how well our YAP/TAZ signature could identify cancer cells with elevated YAP/TAZ activity, we performed GSEA analysis on each of the gene expression datasets analyzed in Figure 7A to compare enrichment of our YAP/TAZ Up and Down genesets with published YAP Up and Down genesets [47,48,49]. Interestingly, despite minimal overlap between these genesets, the Cordenonsi YAP Up, Wang YAP Up, and our YAP/TAZ Up gensets each show significant enrichment in the cells with high YAP and/or TAZ when compared to cells with lower YAP and/or TAZ (Figure 7B,C and Appendix A), indicating that the genes in these genesets are more highly expressed in cells with higher YAP/TAZ activity. Importantly, our YAP/TAZ Up geneset showed similar or greater enrichment (higher NES and lower FDR) than the other existing genesets (Figure 7C and Appendix A). Although our YAP/TAZ Down geneset was negatively enriched in several of the datasets (Figure 7C and Appendix A), the magnitude of this negative enrichment was weaker than the magnitude of the positive enrichment observed for the YAP/TAZ Up genes. This indicates that the downregulated genes are not as consistently YAP/TAZ-dependent as the upregulated genes. The Broad YAP Up and Broad YAP Down genesets did not show strong positive or negative enrichment, respectively, potentially because they were generated from a single cell line (MCF10A), and likely contain genes whose YAP dependence is unique to those cells.

Since our YAP/TAZ signature was enriched in cancer cells with elevated YAP/TAZ activity, we next tested whether it was predictive of YAP/TAZ-TEAD dependence in other cancer cell types, like it was in melanoma. We used Chronos Dependency Scores to identify which of the 1019 cancer cell lines from the DepMap Portal are dependent upon YAP, TAZ, or TEADs for viability. GSEA analysis showed that our YAP/TAZ Up geneset was highly enriched in cancer cell lines that are dependent on YAP/TAZ or TEADs compared to non-dependent cell lines (Figure 7D). To determine the relative enrichment of each geneset in each DepMap cell line, we generated enrichment scores using GSVA analysis (see Appendix A, Tab 4). We then used these GSVA enrichment scores to perform ROC curve analysis, revealing that the enrichment of our YAP/TAZ Up geneset performed similarly or better in predicting YAP/TAZ or TEADs dependence compared to the enrichment of the other YAP Up genesets (Figure 7E). Thus, despite being developed in melanoma cells, the upregulated genes in our YAP/TAZ signature are highly enriched in other cancer cell types with YAP/TAZ-TEAD activation, and the enrichment of this geneset is predictive of cancer cell sensitivity to the loss of YAP/TAZ or TEADs.

## 4. Discussion

### 4.1. A YAP/TAZ Gene Signature That Predicts Dependence upon YAP, TAZ, and TEADs

The experimental and clinical evidence linking YAP or TAZ activation to cancer development and progression suggests that TEAD inhibitors could be effective treatments for several different cancer types, and preclinical studies using TEAD inhibitors have yielded promising results [69,70,71,72]. However, YAP and TAZ are not the only drivers of tumor progression, so TEAD inhibition is only likely to be effective in the subset of patients whose tumors are reliant upon YAP/TAZ/TEADs. Furthermore, predicting sensitivity to TEAD inhibition will be particularly challenging in patients that lack mutations in Hippo Pathway genes. Our results show that YAP/TAZ gene signatures can effectively predict cancer cell dependence upon YAP/TAZ-TEAD, raising the possibility that these signatures could be used to predict a cancer’s sensitivity to TEAD inhibition. Our YAP/TAZ signature could identify cell lines dependent upon YAP/TAZ or TEADs when tested specifically in melanoma or when tested on over 1000 cell lines from different types of cancer. Our data also suggest that YAP/TAZ gene signatures are more effective at predicting dependence upon YAP/TAZ/TEADs than the mRNA levels of *YAP1*, *WWTR1*, or *TEADs*. Indeed, *YAP* and *WWTR1* mRNA expression did not strongly correlate with the expression of known YAP/TAZ target genes (Appendix A) or with dependence upon YAP or TAZ (Appendix A). In contrast, our YAP/TAZ Up geneset could predict dependence upon YAP/TAZ, or TEADs (Figure 5 and Figure 7). Importantly, most of our YAP/TAZ Up genes were also enriched in human melanoma samples with high expression of established YAP/TAZ target genes, suggesting that these genes are also YAP/TAZ-dependent in human melanomas. 

Despite these encouraging results, more work will be necessary to develop diagnostic YAP/TAZ signatures for clinical use. The DepMap dependency data used here are based on cell lines cultured in vitro, and do not capture the complexity of a tumor. Furthermore, human cancers are heterogeneous, which is likely to influence the predictive power of any diagnostic signature. Testing in patient-derived xenograft models could help refine and improve our YAP/TAZ signature, but ultimately, data from clinical trials in patients treated with TEAD inhibitors will likely be necessary to further refine any signature developed using pre-clinical models. Despite the need for more work, our findings suggest that YAP/TAZ-TEAD gene signatures have potential diagnostic value that warrants further development. Since patients with, or at risk of developing, distant metastasis have the greatest need for more effective treatments, it will be important to understand if the site of metastasis influences sensitivity to YAP/TAZ-TEAD inhibition. It will also be important to determine whether the site of metastasis influences how accurately a YAP/TAZ signature predicts sensitivity to YAP/TAZ-TEAD inhibition. Here we found that YAP promotes lung metastasis formation by melanoma cells, but we did not investigate the roles of YAP or TAZ in melanoma metastasis to other organs. Other preliminary data from our lab have revealed that the knockdown of YAP and TAZ in highly metastatic human melanoma cells impaired disseminated tumor cell seeding of the liver, brain, kidneys, and spleen, as well as metastasis formation in the brain, liver, and bone following intracardiac injection (not shown). While additional work is needed to confirm these preliminary findings, overall, these results suggest that YAP/TAZ-TEAD activity impacts melanoma metastasis to multiple organ sites. 

The fact that our YAP/TAZ Up geneset was predictive when tested on a diverse set of cancer cell lines (Figure 7E) suggests that it may not be necessary to develop cancer type-specific YAP/TAZ signatures. However, some of our findings suggest otherwise. We found that several of our YAP/TAZ signature genes that were highly conserved in melanoma cell lines and human melanomas were not YAP/TAZ-responsive in many other cancer types that we analyzed (Figure 7A). This suggests that these genes may be more reliable readouts for YAP/TAZ-TEAD activity in melanoma than in other cancers. Furthermore, our YAP/TAZ Up geneset, which was derived using melanoma cell lines, showed stronger enrichment in YAP/TAZ and TEAD-dependent melanoma cell lines than the other published YAP Up genesets that were derived from other cell types (Figure 5A). We also found that although some established YAP/TAZ target genes (*CRIM1*, *F3*, *ANKRD1*) correlated strongly with *CTGF* and *CYR61* expression in human melanoma, others did not (*WWC1*, *FOSL1*, *FSTL3*) (Figure 6). Collectively, this suggests that while YAP/TAZ signatures may generally have predictive value across multiple cancer types, cancer type-specific YAP/TAZ signatures may provide an added level of accuracy for that cancer type.

Although our findings showed that increased expression of YAP/TAZ target genes, which is likely due to elevated YAP/TAZ activity, was a reliable indicator of dependence upon YAP/TAZ-TEAD, there were some cell lines that did not appear to follow this pattern. For example, WM793 cells showed high expression of our YAP/TAZ signature genes, but were not sensitive to loss of YAP, TAZ, or TEADs, while NZM42 and CHL1DM cells, which were sensitive to loss of TAZ, had relatively low expression of YAP/TAZ signature genes (Figure 5). The DepMap dependency data are based on the CRISPR-mediated knockout of individual genes, so there could be compensation of YAP or TAZ for each other. TEADs could also compensate for each another. Although it is beyond the scope and feasibility of this study, testing a larger cohort of cell lines for sensitivity to pan-TEAD inhibitors could help refine our YAP/TAZ signature to enhance its predictive accuracy. 

The development of therapeutic resistance is common in patients treated with targeted therapies, and the upregulation of YAP/TAZ-TEAD transcriptional activity has been found to be a resistance mechanism in cancer cells treated with targeted therapies [73,74]. For example, YAP/TAZ activation plays a causal role in resistance to KRAS, BRAF, MEK and EGFR inhibitors [44,72,75,76,77,78,79,80,81,82]. Collectively, this suggests that TEAD inhibitors could be effective second-line therapies for some patients, and that YAP/TAZ-TEAD signatures, like ours, might provide a means to identify which patients would benefit. Whether used as primary or secondary therapies, we should anticipate that some patients will develop resistance to TEAD inhibitors. Transcriptional profiling in a large cohort of cancer cell lines treated with pan-TEAD inhibitors could help predict and prevent such resistance. Consistently, a recent study using a panel of cancer cell lines found that treatments with the TEAD inhibitor MGH-CP1 and YAP/TAZ silencing both promoted the VGLL3-mediated transcriptional activation of SOX4/PI3K/AKT signaling, which contributed to resistance to MGH-CP1 [83]. Importantly, this study further demonstrated that the dual inhibition of AKT and TEAD could help overcome this resistance.

### 4.2. The Complexity of YAP/TAZ-TEAD-Dependent Gene Expression in Cancer Cells

Several of our findings highlight the complexity of YAP/TAZ-TEAD-dependent gene expression programs. We found minimal overlap in the differentially expressed genes in each of the melanoma cell lines we analyzed (Figure 2). The genes that did overlap in melanoma cells were all also YAP/TAZ-dependent in some other cancer cell lines (Figure 7A), but the individual genes that responded to changes in YAP/TAZ activity in each cell line varied dramatically. Even well-established target genes like *CTGF*, *CYR61*, *ANKRD1*, and *WWC1* appeared to be YAP/TAZ-independent in some cell lines. While some of the non-overlapping genes in the melanoma cell lines that we analyzed may be indirect targets, the majority of the genes in our YAP/TAZ signature had TEAD peaks in publicly available ChIP-Seq datasets (Figure 4D), indicating that these could be direct YAP/TAZ-TEAD target genes. This suggests that contextual factors beyond the presence or absence of TEAD motifs in a gene’s regulatory elements influence YAP/TAZ-TEAD-dependent gene expression programs. Indeed, both YAP and TAZ can bind chromatin remodeling complexes and influence chromatin accessibility [84,85,86]. YAP and TAZ also bind a long and growing list of transcription factors and proteins that can dramatically influence the transcriptional landscapes of cells and tissues [18,84,87].

It is also important to note that the data we analyzed here do not consider temporal differences in YAP/TAZ-dependent gene expression. Indeed, a recent study nicely demonstrated that the regions of the genome occupied by TAZ change in a temporal manner following TAZ activation [88]. Our data also suggest that the magnitude of YAP or TAZ activation will influence which potential YAP/TAZ-TEAD target genes are changing in a substantial manner. For example, the same genes were influenced by YAP^2SA^ and YAP^S127A^ in A375 cells. However, in A375-YAP^S127A^ cells, which have much lower YAP/TAZ-TEAD activity, only a fraction of those genes changed in a significant (greater than 2-fold) manner (Figure 1F,G). This suggests that as more YAP or TAZ enters the nucleus and binds to TEADs, a greater number of target genes will significantly change, which highlights the need to ensure that experimental manipulation of YAP/TAZ-TEAD activity mimics biologically relevant changes in activity that occur during disease processes. Similarly, since cell density, cell geometry, and substrate rigidity each substantially impact YAP and TAZ nuclear localization and activity [89], these critical variables will also influence which subsets of target genes are changing in a significant manner, and potentially which tumors will respond to TEAD inhibition. While we performed our experiments on cultures of similar cell density, we also used publicly available gene expression data from other labs, so we cannot rule out the possibility that disparate culture conditions are influencing YAP/TAZ target genes in these datasets. 

Another interesting observation from our study is that although the numbers of genes upregulated and downregulated by YAP and TAZ were generally similar in the cell lines we analyzed, the upregulated genes were more highly conserved across different cancer types. Upregulated genes were also better predictors of YAP/TAZ and TEAD dependence (Figure 5 and Figure 7), and strongly correlated with other known YAP/TAZ target genes in human patient samples (Figure 6). In contrast, downregulated genes did not show the expected inverse correlation with known YAP/TAZ target genes in human patient samples (Appendix A). Most of the downregulated genes did have TEAD peaks in available ChIP-Seq datasets (Figure 4D), but the presence of these peaks was much less consistent across the datasets than it was for the upregulated genes. It is unclear why this is the case, but perhaps in the context of cancer, where YAP-TEAD and TAZ-TEAD tend to be more active, they are more likely to promote gene expression than to inhibit it. 

While YAP/TAZ gene signatures offer potential diagnostic power for sensitivity to TEAD inhibition, individual YAP/TAZ target genes may also have therapeutic potential. Indeed, some of the genes in our signature have already been reported as drivers of YAP/TAZ-mediated tumor progression and metastasis. For example, YAP-mediated melanoma cell invasion was found to require the YAP-dependent induction of *AXL*, *CRIM1*, and *CYR61* [6]. The TAZ-mediated induction of *AXL* was also found to drive an AXL–ABL2–TAZ feed-forward loop that promoted lung adenocarcinoma brain metastasis [65]. Another study found that YAP promoted breast cancer metastasis through the induction of the *ITGB2* gene [90]. *NUAK2* is a YAP/TAZ target gene that promotes tumor progression in liver and bladder cancer [91,92], and *CTGF*, a well-established YAP/TAZ target gene, has clear roles in cancer [93]. Several other genes in our signature have also been implicated in cancer, but their roles in YAP/TAZ-mediated tumor progression and metastasis have not yet been explored. Although determining which YAP/TAZ target genes are required for tumor progression and metastasis could reveal potential therapeutic targets, this is beyond the scope of this study, and could prove challenging if multiple genes with overlapping functions are involved.

## 5. Conclusions

With TEAD inhibitors already in clinical trials, the need for diagnostic tests that can identify patients who are likely to benefit from these drugs is essential. Since YAP and TAZ are extensively regulated at the posttranslational level through complex signaling networks, protein and mRNA expression levels of YAP or TAZ are not likely to be a reliable readout for YAP/TAZ-TEAD activity. Our study suggests that YAP/TAZ-TEAD gene signatures may provide a more reliable means to assess YAP/TAZ-TEAD activation and, potentially, help predict sensitivity to TEAD inhibition. The YAP/TAZ signature we developed here was predictive in melanoma cell lines and more broadly in a large set of cancer cell lines. The genes in this signature are TEAD-dependent, and thus a direct readout for TEAD-dependent gene expression. Our work raises the intriguing possibility that with additional testing and refinement using pre-clinical models and patient data, our YAP/TAZ signature could be used as a diagnostic test to help identify patients likely to respond favorably to TEAD inhibitors. Although we did not explore this in the current study, we hypothesize that since the genes in this signature are regulated by YAP/TAZ-TEAD, they may also have value as biomarkers to monitor the efficacy of TEAD inhibition during treatment with TEAD inhibitors or other drugs targeting the YAP/TAZ-TEAD axis. 

## Figures and Tables

**Figure 1 cancers-16-00852-f001:**
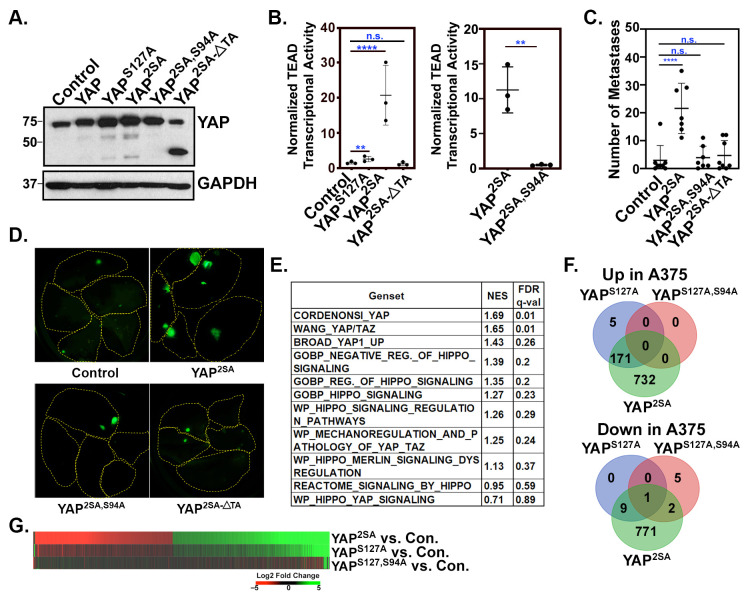
YAP promotes melanoma metastasis in a TEAD- and transactivation domain-dependent manner. GFP-expressing A375 cells were stably transduced with a control empty vector (MSCV-IRES-Hygro) or the indicated YAP constructs and then assayed by (**A**) Western blot or for (**B**) TEAD transcriptional activity using a dual-luciferase reporter assay. (**C**) Cells from (**A**) were injected into NOD/SCID mice via the lateral tail vein and after 19 days, the numbers of GFP-positive metastases were counted in the lungs. (**D**) Fluorescent images of all 5 lung lobes from 1 representative animal of each group in panel (**C**). Lung lobes are outlined in each image. (**F**) RNA-seq was performed on 3 independent RNA samples from A375 cells expressing an empty vector control (MSCV-IRES-Hygro) or the indicated YAP constructs, and the number of genes significantly up- or downregulated (fold change > 2, adjusted *p* Value < 0.05) by each YAP construct relative to the control is shown. (**E**) Geneset enrichment analysis (GSEA) was run on the RNA-seq data from (**D**) to test for the enrichment of the indicated Hippo Pathway genesets in the A375-YAP^2SA^ vs. A375-Control cells. The normalized enrichment score (NES) and false discovery rate (FDR) q Value are listed for each geneset. (**G**) The heatmap shows log2 fold change (log2FC) for each indicated comparison for all 1696 genes that were differentially expressed in the A375-YAP^2SA^ vs. Control cells. The genes are arranged from most upregulated (green) to most downregulated (red) in the A375-YAP^2SA^ cells. Data used to generate this heatmap can be found in Appendix A. The plots in (**B**,**C**) show mean ± SD with each dot representing an *n*. *n* = 3 independent experiments in (**B**) and *n* = individual mice in (**C**). Statistical significance was determined using one-way ANOVA with Dunnett’s multiple comparisons test (**B**, **left** and **C**) or unpaired, two-tail *t*-test (**B**, **right**); ** *p* ≤ 0.01, **** *p* ≤ 0.0001, n.s. *p* > 0.05.

**Figure 2 cancers-16-00852-f002:**
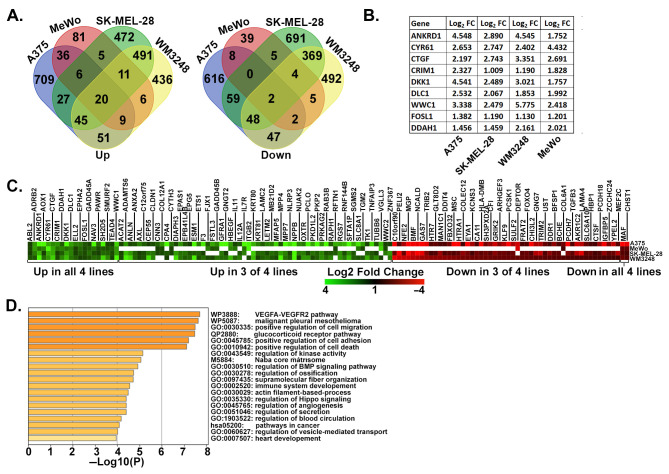
Identification of a YAP/TAZ gene signature in metastatic human melanoma cells. (**A**) Venn diagrams show overlap between genes upregulated (**left**) or downregulated (**right**) (≥2 fold, *p* Value < 0.05) in the following: A375-YAP^2SA^ vs. Control cells (our data); MeWo YAP^5SA^ vs. Control cells ([6]); Control siRNA vs. YAP/TAZ siRNA transfected SK-MEL-28 or WM3248 cells (GSE68599). (**B**) The table shows fold change for established YAP/TAZ target genes. All changes are statistically significant (adjusted *p* Value *p* ≤ 0.001). (**C**) The heatmap shows fold change for the 132 YAP/TAZ dependent genes in our YAP/TAZ signature. White indicates the gene was not detected in A375, WM3248, and SK-MEL-28, or not differentially expressed in MeWo. The data used to generate this heatmap can be found in Appendix A, Tab 6. (**D**) Pathway analysis was performed with the YAP/TAZ Up geneset using Metascape [38]. The top 20 enriched clusters are shown.

**Figure 3 cancers-16-00852-f003:**
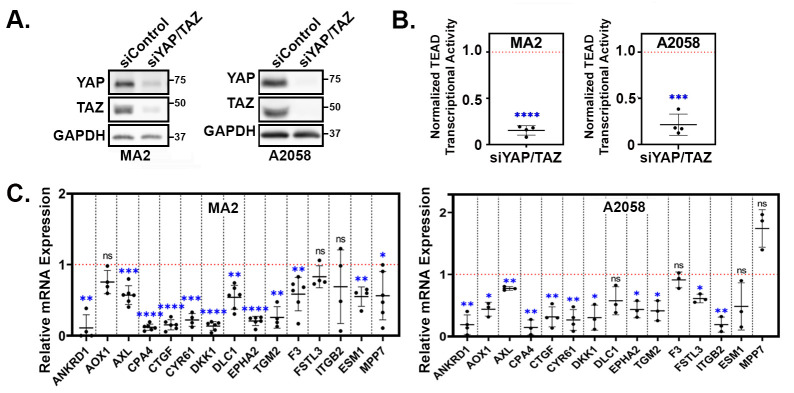
YAP/TAZ signature genes are YAP/TAZ-dependent in metastatic melanoma cells. (**A**–**C**) A375-MA2 or A2058 cells were transfected with either a control siRNA or combined siRNA SMARTpools targeting YAP and TAZ for 24 h. Cells were then trypsinized and replated for an additional 48 h and then assayed by (**A**) Western blot (**B**) for TEAD transcriptional activity using a dual-luciferase reporter assay, or by (**C**) qPCR for the indicated genes. The plots (**B**,**C**) show the fold change in the siYAP/TAZ samples compared to the siControl samples, which were set to 1, and represented as a red dotted line. Each data point is an independent experiment and the mean ± SD is shown. Statistical significance was tested using a one-sample *t*-test; * *p* ≤ 0.05, ** *p* ≤ 0.01, *** *p* ≤ 0.001, **** *p* ≤ 0.0001, n.s. *p* > 0.05.

**Figure 4 cancers-16-00852-f004:**
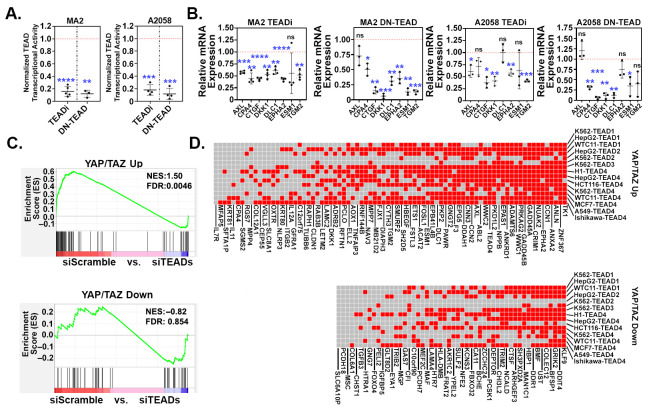
YAP/TAZ signature genes are TEAD-dependent. (**A**,**B**) A375-MA2 or A2058 cells stably transduced with control vector, TEADi, or TEAD1^Y421E^ (DN-TEAD1) were assayed for (**A)** TEAD transcriptional activity using a dual-luciferase reporter assay or (**B**) by qPCR for the indicated genes. The plots (**A**,**B**) show the fold change in the TEADi or DN-TEAD samples compared to the control samples, which were set to 1 and represented as a red dotted line. Each data point is an independent experiment and the mean ± SD is shown. Statistical significance was tested using a one-sample *t*-test; * *p* ≤ 0.05, ** *p* ≤ 0.01, *** *p* ≤ 0.001, **** *p* ≤ 0.0001, n.s. *p* > 0.05. (**C**) Gene Set Enrichment Analysis (GSEA) was performed using our YAP/TAZ Up and YAP/TAZ Down genesets and a publicly available dataset (GSE60664, [46]) in which all 4 TEADs were knocked down in human melanoma cells (MM047). Shown are the enrichment plots with Normalized Enrichment Score (NES) and False Discovery Rate (FDR) for each geneset. (**D**) ChIP-seq datasets with the indicated TEADs were downloaded from ENCODE and analyzed for the presence of peaks associated with each YAP/TAZ signature gene. The heatmap indicates which genes had a TEAD peak for each dataset. Red indicates at least 1 TEAD peak mapped to that gene, gray indicates no TEAD peaks mapped to that gene.

**Figure 5 cancers-16-00852-f005:**
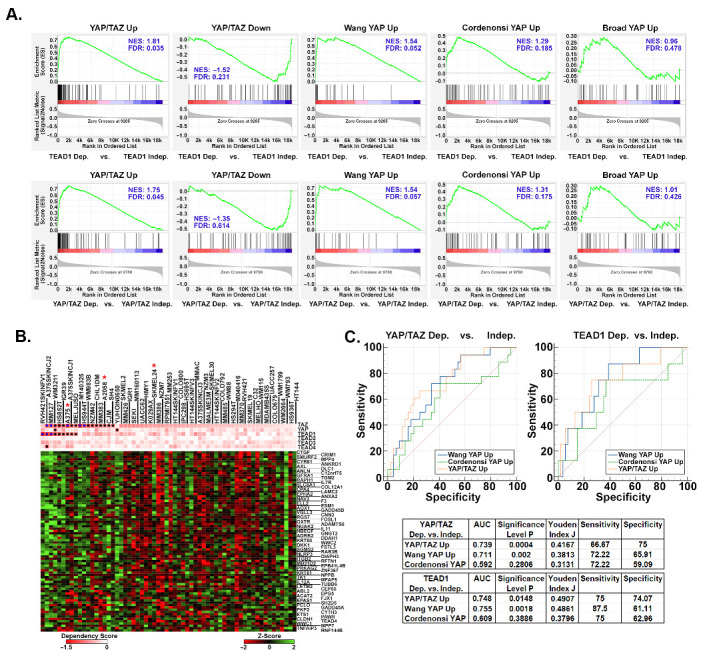
Upregulated YAP/TAZ signature genes are predictive of melanoma cell dependence upon YAP/TAZ-TEAD. RNA-seq data and dependency scores for 62 melanoma cell lines were downloaded from the DepMap Portal (Appendix A). Cell lines were scored as dependent (Chronos Dependency Score of ≤−0.65) or independent (Chronos Dependency Score of ≥−0.65) for TEADs 1–4, YAP, or TAZ (WWTR1). (**A**) GSEA was performed on RNA-seq data to test for enrichment of the indicated genesets in cell lines dependent upon TEADs or either YAP or TAZ (YAP/TAZ). (**B**) The heatmap shows the relative expression (Z-Score of the log transformed TPM (log2(1 + TPM))) of each of the 80 YAP/TAZ Up genes in the DepMap melanoma cell lines. Chronos Dependency Scores for TEADs, YAP, or TAZ are shown in the pink and white heatmap. Blue dots indicate cell lines with Chronos Dependency Scores ≤−1.0, black dots indicate a score between −1.0 and −0.65. A red asterisk (*) indicates that the melanoma cell line was analyzed above for expression of the genes in our signature. (**C**) GSVA was used to score the enrichment of each indicated geneset in each of the 62 melanoma cell lines in (**B**) and then ROC curves were generated to test how well GSVA score could predict dependency upon YAP/TAZ or TEADs. The area under the curve (AUC), *p* Value, Youden Index J, Sensitivity, and Specificity values for each geneset are shown in the tables. The data used to generate this figure can be found in Appendix A.

**Figure 6 cancers-16-00852-f006:**
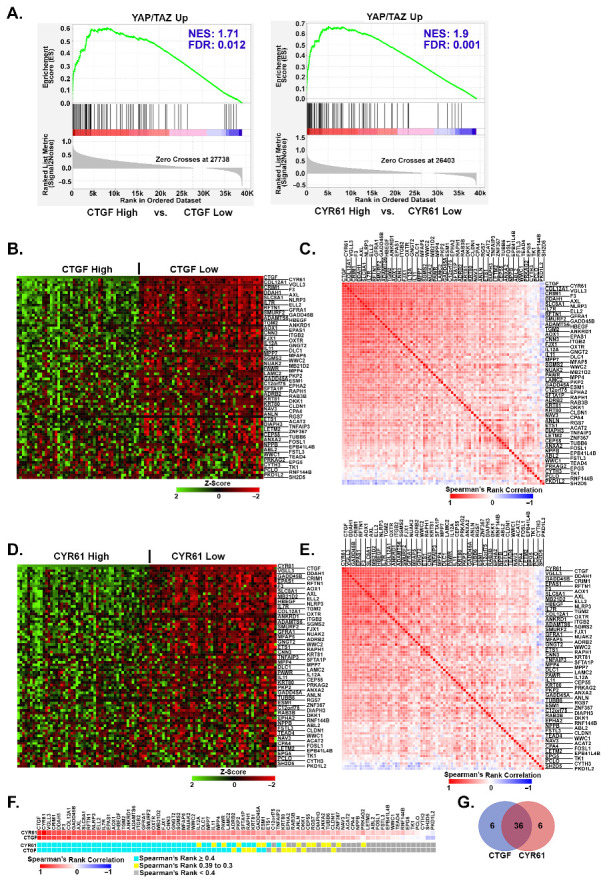
Upregulated YAP/TAZ signature genes are correlated with increased YAP/TAZ activity in human melanoma. RNA-seq data from the TCGA human skin cutaneous melanoma (SKCM) project were downloaded and tumors with high (≥1 standard deviation from the mean) or low (≤−1 standard deviation from the mean) expression of *CTGF* or *CYR61* mRNA were analyzed. (**A**) GSEA analysis was performed to test for the enrichment of the YAP/TAZ Up geneset in *CTGF* or *CYR61* high vs. low tumors (NES and FDR are indicated). (**B**,**D**) The heatmaps show the relative expression (Z-Score of the log transformed TPM (log2(1 + TPM))) of each of the 80 YAP/TAZ Up genes in *CTGF* (**B**) or *CYR61* (**D**) high vs. low tumors. Tumors are sorted by *CTGF* or *CYR61* mRNA expression and genes are ranked from highest (**top**) to lowest (**bottom**) based on their Spearman Rank Correlation with either *CTGF* (**B**) or *CYR61* (**D**). (**C**,**E**) Spearman similarity matrix analysis was performed on the expression data shown in (**B**) and (**D**) and the Spearman Rank Correlation value for each pairwise comparison across all tumors analyzed in each set is shown. Genes are ranked the same as in (**B**,**D**). (**F**) The Spearman Rank Correlation values for each gene compared to *CTGF* or *CYR61* (same data as 1st rows in (**C**,**E**)). Genes with Correlation values ≥0.4 are indicated in blue, between 0.39 and 0.3 in yellow and <0.3 in gray. (**G**) The Venn diagram shows the number of genes from each comparison in (**F**) with correlation values ≥0.4. The data used to generate this figure can be found in Appendix A.

**Figure 7 cancers-16-00852-f007:**
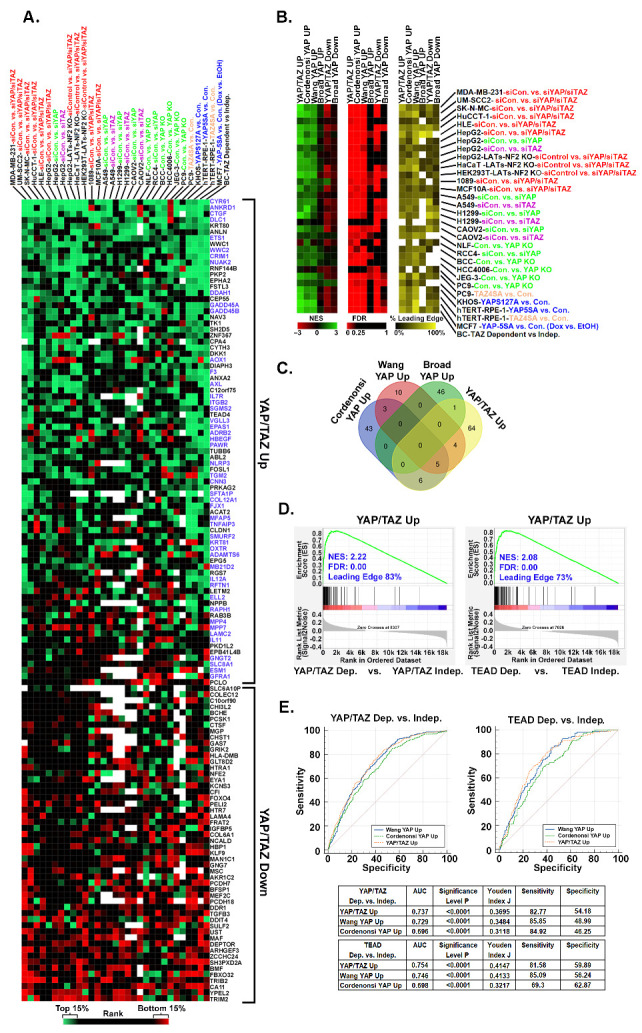
YAP/TAZ signature genes are YAP/TAZ-dependent and predictive of dependence upon YAP/TAZ-TEAD in other cancer types. Publicly available datasets (Appendix A, Tab 3) were downloaded from NCBI-GEO and GSEA was used to generate rank-ordered lists for the indicated cell lines and comparisons. (**A**) The heatmap shows the % Rank (the gene’s rank/total genes in rank-ordered list × 100) of each of the 80 YAP/TAZ Up and 52 YAP/TAZ Down genes. Green indicates that the gene’s rank is between 0 and 15% (i.e., enriched in YAP and/or TAZ high cells), red between 85 and 100% (i.e., enriched in the YAP and/or TAZ low cells), black between 15 and 85% (i.e., not enriched), and white indicates the gene was not included in the rank-ordered list for that dataset. (**B**) GSEA was performed on the same publicly available datasets using our YAP/TAZ Up and Down genesets and other published YAP Up genesets. The heatmaps show the Normalized Enrichment Score (NES), False Discovery Rate (FDR), and % of each gene set that was in the Leading Edge (% Leading Edge) for each comparison. (**C**) The Venn diagram shows overlap between our YAP/TAZ Up geneset and other published YAP Up genesets. Lists of each geneset are found in Appendix A, Tab 4, GSEA results, and the % Rank values used to generate the heatmaps in this figure are found in Appendix A. (**D**) RNA-seq data and dependency scores for all 1019 cell lines were downloaded from the DepMap Portal (see Appendix A). Cell lines were scored as dependent (Chronos Dependency Score of ≤−0.65) or independent (Chronos Dependency Score of ≥−0.65) for TEADs 1–4, YAP, or TAZ. GSEA was performed on the RNA-seq data to test for enrichment of our YAP/TAZ Up genset. (**E**) GSVA was used to score the enrichment of each indicated geneset in each of the 1019 cell lines in the DepMap Portal and then ROC curves were generated to test if the GSVA score could predict dependency upon YAP/TAZ or TEADs. Area Under the Curve, *p* Value, Youden Index J, Sensitivity, and Specificity values for each geneset are shown in the tables.

## Data Availability

The RNA-seq data generated for this study are openly available in NCBI GEO accession number GSE234083. All third party data used in these analyses are available in the publicly accessible repositories listed in either the Appendix A.

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
