# Peer review of "Identification of a Gene Signature That Predicts Dependence upon YAP/TAZ-TEAD"

_cancers, 2024, doi:10.3390/cancers16050852_

Round 1

Reviewer 1 Report

Comments and Suggestions for Authors

Ryan et. al. developed a YAPTAZ gene signature from metastatic human melanoma cells that has diagnostic value for identifying cancer patients who will benefit from TEAD inhibitors. This signature is predictive of cancer cell dependence on YAP, TAZ, and TEAD, and it could be used as a diagnostic test to identify patients likely to respond favorably to TEAD inhibitors. 

In general, I think the authors did some work, and the results are convincing and interesting. The analysis pipeline that the authors used is sophisticated and impeccable. Although some details need further explanation, I have some questions related to the methods and results. I have no major criticisms. Here are some questions and comments that the authors should think about as they make changes to their work.

1.     Accurate experimental processing and analysis techniques are necessary for cell lines since the identification of the YAP/TAZ gene signature is dependent on related studies on cell lines. In the Materials and Methods section's description of cell line culture, I saw that the author omitted important details about the culture procedure. It only enumerates the culture's external environmental conditions and the reagents utilized. This is insufficient. Tao Li and colleagues studied HeLa cells and discovered that DLG2, a crucial component of the YAP/TAZ axis, is expressed differently in various cell subpopulations subject to density-dependent selection. This variation also impacts the phosphorylation process of YAP/TAZ (PMC8288455). One important step in the activation of YAP/TAZ is phosphorylation. Because of this, the author is unsure of how the cell density throughout the cell culture phase impacted the differential expression study that followed. The Materials and Methods section omits information regarding the author's cell state and additional parameters used while extracting RNA for RNA_Seq. Based on this, I think the Materials and Methods section has to provide a detailed explanation of these crucial cell culture process characteristics, like the number of cells, the subculture cycle, the time it takes for the cells to double, the time point for RNA extraction, etc. Similarly, this component should also be explored in the Discussion section since YAP/TAZ activation is affected by density-dependent selection. For instance, if solid tumors and non-solid tumors can be treated with the same targeted TEAD inhibitors, etc.

2.     The resolution of the Figures is really low. Numerous significant annotations are difficult to see. In the updated edition, the writers must present clear images.

3.     What does the band around marker 50 in Figure 1A represent? The data in the two subfigures of Figure 1B that belong to the same YAP2SA are inconsistent. Why does this occur? How are the transfer sites counted in the associated results that correspond to Figure 1C? The usage of a fluorescent stereomicroscope was stated by the author, but no detailed pre-processing steps, such as the type of transparency processing method employed, were provided. Moreover, why limit attention to lung metastases alone? Are there also liver metastases? If liver metastasis was utilized as a proxy for metastasis, would the outcomes differ?

4.     The control group for statistical testing is absent from Figures 3B and C.

5.     The author adduces in lines 305–307 that alternative experimental methods to those in Result 3.1 were conducted in various melanoma cells to demonstrate that the YAP/TAZ gene signature that the author discovered is universally TEAD-dependent. The A375 cell line must undergo the same experimental procedure. We are unable to reach this conclusion unless the experimental data are consistent.

6.     According to the authors, there is a "stronger" association between WWTR1 mRNA expression and line 385. The identical text can be found on line 388. Here, what does "stronger" mean? There is insufficient statistical test data, as indicated by Figure S2, to support the reasonableness of this phrasing.

7.     Only ten genes were examined in Figure 2B's examination of target gene expression variations between the CTGF gene's high and low expression groups. But 11 genes were included in previous analyses? What criteria are used to classify high and low expressions?

8.     The author feels that CYR61 and CTGF are more reliable markers of YAP/TAZ activation than YAP/TAZ mRNA expression level, based on the description provided in lines 395–397. This is a clear conclusion. The transcription products of YAP/TAZ upon phosphorylation and nuclear entry include CTGF and CYR61. YAP/TAZ phosphorylation indicates that the protein has been activated. There is no need to demonstrate this using the analysis of Figure 2.

9.     On line 464, the author lists 1019 cell lines. What precisely are they? I could not locate the information in the paper. Moreover, are the 1019 cell lines comprised of both solid and non-solid tumor cell lines?

10.  Although there are only 72 references in the manuscript, there are 163 references in the reference list at the end of the article. Why does this occur?

Reviewer 2 Report

Comments and Suggestions for Authors

The authors provide an extensive study on the hippo pathway in multiple cancer types, primarily focusing on melanoma and extending to other types of cancer. They used multiple publicly available datasets and analyzed the data sets concerning hippo pathway targets. The study is essential regarding understanding this critical pathway and finding druggable targets. Some of the comments are below:

It will be better to list the genes that the authors think are important for predicting the activity in a table in the main text and possibly provide a score for total activity since the detailed analysis has somewhat obscured the gene list or are they proposing the entirety of the GCT file used for the GSEA analysis? They could contrast this with the known/verified/predicted TEAD binding gens such as https://maayanlab.cloud/Harmonizome/gene_set/TEAD1/. This could be made clear by the authors in the text.

Did the authors check the localization of the mutated YAPs? Could they mention/cite previous publications? In terms of interaction, has this been shown before? Also, please check lines 256-57. Which one is the insensitive form? Double-mutated one? Could the author also comment on the difference between the two mutations since RNA seq data show interaction is probably more with 25A? Figure 1E, has this been sorted in some way? The heatmap shows a rescue of the gene expression pattern in the double-mutant. How do the authors interpret this?

For Figure 2, did the author check YAP/TAZ downregulation in A375? Why was a different cell type transfected with siRNA? This is also true, for example, in Figure 4. In this case, they are using two other cell lines that were not used previously. Could the author include an explanation for their choice of cell lines?

Can the authors include how the gct for yaz/tead up and down were created or included as supplementary? Or is this the list in Supplementary File 3? Could the author describe the process in the methodology section since this is the most crucial aspect of the work?

in their KD or overexpression RNAseq did they observe the changes in the hippo pathway? Or can they do GSEA on the data set directly?

 Line 220: could you please clarify? Is this the promoter?

Figure 5B, Can the cell lines used in the study be pointed out/marked?

Figure 7D: the TEADdep and YAP/TAZ dependent signatures. How much is the overlap in the gene sets? Do all of these genes have TEAD CHipSeq peaks?

Why was tophat-cufflink used instead of salmon/kallisto/deseq2 or STAR/rsem/Deseq2? The author may want to consider those in the future, along with limma-voom etc.

Please include the markers in the blue/X-ray film gel regarding the original western. 

Author Response

REVIEWER 2:

Response: We thank Reviewer 2 for taking time to review our work and for their helpful suggestions. We have addressed their comments and questions below and have revised the manuscript in response to their helpful suggestions.

R2 Comment #1:   It will be better to list the genes that the authors think are important for predicting the activity in a table in the main text and possibly provide a score for total activity since the detailed analysis has somewhat obscured the gene list or are they proposing the entirety of the GCT file used for the GSEA analysis? They could contrast this with the known/verified/predicted TEAD binding gens such as https://maayanlab.cloud/Harmonizome/gene_set/TEAD1/. This could be made clear by the authors in the text.

Response: The genes in the YAP/TAZ Up and YAP/TAZ Down genesets that we identified are shown Figure 2C and listed in Supplemental Table 1. We uploaded the supplemental tables with our submission but do not know if reviewers had access to this data. The entire YAP/TAZ Up geneset (80 genes) was used for all the analyses in this manuscript including the analysis of DepMap to test the predictability of this geneset. We clarified this in the text as suggested. Our data shows that enrichment of the entire YAP/TAZ Up geneset (all 80 genes) was predictive in both melanoma cell lines (Figure 5C) and more broadly across all the cancer lines we analyzed (Figure 7E). To determine relative enrichment of the entire YAP/TAZ Up geneset in each cell line in the DepMap dataset, we used GSVA. This score was used to generate the ROC curves in Figure 5C and 7E.  As suggested, we now included the GSVA value for each melanoma cell line in Tables S4 and for all 1019 cell lines analyzed in Table S6.  We have also amended the text in the Results section and updated the appropriate Figure Legends. Our analysis of several existing ChIP-seq datasets in Figure 4 is similar to the reviewer’s suggestion to cross-reference these genes with known/verified/predicted TEAD binding genes and shows that TEADs bind to the genomic regions of all but 1 of the genes in our YAP/TAZ Up gene set.

R2 Comment #2:  Did the authors check the localization of the mutated YAPs? Could they mention/cite previous publications? In terms of interaction, has this been shown before? Also, please check lines 256-57. Which one is the insensitive form? Double-mutated one? Could the author also comment on the difference between the two mutations since RNA seq data show interaction is probably more with 25A? Figure 1E, has this been sorted in some way? The heatmap shows a rescue of the gene expression pattern in the double-mutant. How do the authors interpret this?

Response: As requested, we have added citations for the initial papers describing the LATS-insensitive and TEAD binding mutants of YAP utilized here. These mutations have been extensively utilized and characterized in the field. Collectively, this works shows that LATS-mediated phosphorylation of Serine 127 promotes cytoplasmic sequestration, while phosphorylation on Serine 397 (which is Serine 381 in the YAP isoform #3, which we use in this work) promotes proteasomal degradation. The YAPS127A mutant is insensitive to LATS-mediated cytoplasmic sequestration and the YAP2SA mutant is insensitive to both LATS-mediated cytoplasmic sequestration and LATS-mediated proteasomal degradation [1,2]. We now more clearly describe these mutants in this section of the results.

In Figure 1E (now Figure 1G in the revised manuscript), the heatmap shows all genes that are differentially expressed in the YAP2SA-expressing A375 cells relative to the control vector expressing cells (the names of these genes, their fold change, and the adjusted pValues are listed in Table S2). The genes are sorted based on fold change from most positive (green) to the most negative (red). The fold change for the same genes in the YAPS127A and YAPS127,S94A  -expressing A735 cells are also shown in Figure 1G and their fold changes and adjusted pValues are also listed in Table S2. We have added these details to the Figure Legend. The heatmap shows that the majority of the genes that are differentially expressed in the A375-YAP2SA cells are also changing in the same direction in the A375-YAPS127A cells, but to a lesser magnitude, which we suggest is because the activity of the YAPS127A mutant is lower than that of the YAP2SA mutant (see Figure 1B). The fact that very few of the genes change in the A375 cells expressing the YAPS127A,S94A mutant, which cannot bind TEADs, demonstrates that YAP requires TEAD binding to influence the expression of the vast majority of these genes.

R2 Comment #3: For Figure 2, did the author check YAP/TAZ downregulation in A375? Why was a different cell type transfected with siRNA? This is also true, for example, in Figure 4. In this case, they are using two other cell lines that were not used previously. Could the author include an explanation for their choice of cell lines?

Response: We have not tested the expression of the indicated genes in A375 cells transfected with YAP/TAZ siRNAs because the data in Figure 1 already demonstrates YAP-mediated regulation of these genes in A375 cells. The cell lines in Figures 1-4 were selected because they are each metastatic human melanoma cell lines, and our goal was to identify a set of genes that is regulated by YAP/TAZ in metastatic melanoma cells. In Figure 2 we identified YAP/TAZ target genes using global gene expression datasets from 4 metastatic human melanoma cell lines to identify a YAP/TAZ signature. For added rigor, we then used qPCR to test if a subset of those genes is also YAP/TAZ-dependent in two additional metastatic melanoma cell lines (MA2 and A2058) in Figure 3.  We chose the MA2 and A2058 cell lines for this because another study in our lab demonstrated that knockdown of YAP and TAZ in these cell lines impaired metastasis formation in mice. The rationale for using these cell lines is now included in the text.

R2 Comment #4: Can the authors include how the gct for yaz/tead up and down were created or included as supplementary? Or is this the list in Supplementary File 3? Could the author describe the process in the methodology section since this is the most crucial aspect of the work?

Response: The information the reviewer requested is included in Table S3. We uploaded all supplemental material with our submission and apologize if this was not provided to the reviewers. Briefly, in Figure 2 we identified 132 genes that are YAP/TAZ dependent in at least 3 of these 4 metastatic melanoma cell lines. This YAP/TAZ signature is shown in Figure 2C with each upregulated and downregulated gene’s name listed in the heatmap. The Venn diagram in Figure 2A shows the numbers of differentially expressed genes in each cell line and their overlap. Table S3, Tabs 1-4 provide the lists of all genes that are differentially expressed in each cell line with their fold change and pValues. Separate tabs showing the overlap (Tab 5) and just the 132 YAP/TAZ genes in the YAP/TAZ signature (Table S3, Tab 6) are also included. The Figure Legend for this table describes how the differentially expressed genes were identified, but as suggested we added these details to the Methods Section.

R2 Comment #5: in their KD or overexpression RNAseq did they observe the changes in the hippo pathway? Or can they do GSEA on the data set directly?

Response: It is well established that YAP/TAZ can regulate the expression of several genes that encode proteins in the Hippo Pathway or that regulate the Hippo Pathway. Consistently, we found that several Hippo Pathway genes were among the DEG in our RNAseq data. To address this comment more directly, we ran GSEA on our RNAseq dataset using Molecular Signature Database genesets associated with the Hippo Pathway or Hippo Pathway signaling (see new Figure 1E). We found that, as expected, the genesets associated with increased YAP/TAZ signaling were highly enriched in the A375-YAP2SA cells. Some of the Hippo Signaling pathway genesets were also enriched, likely because, as noted above, YAP and TAZ regulate the expression of several genes that encode proteins in the Hippo Pathway. We have revised Figure 1, its Legend, the Results section, and the Supplemental Materials to include this new data.

R2 Comment #6:  Line 220: could you please clarify? Is this the promoter?

Response: For the analysis of ChIP-seq data in Figure 4 we used the ChIPseeker package in Rm, a standard package for peak annotation in ChIP-Seq data analysis pipelines. In ChIPseeker, the parameter used to determine whether a peak is annotated to the promoter region of a particular gene is the “tssRegion” parameter (i.e. transcriptional start site region), which estimates the promoter region for a given gene based on a certain genomic distance up and downstream of the gene’s transcriptional start site. We used a conservative +/- 1kb from TSS to define the promoter regions of genes. We edited the Methods to better clarify parameters in the ChIP analysis pipeline.

R2 Comment #7: Figure 5B, Can the cell lines used in the study be pointed out/marked?

Response: The DepMap database does not include all the cell lines used in this study, but we have placed a red asterisk next to cell lines that are included in Figure 5b and updated the Figure Legend.  

R2 Comment #8: Figure 7D: the TEADdep and YAP/TAZ dependent signatures. How much is the overlap in the gene sets? Do all of these genes have TEAD CHipSeq peaks?

Response: Of the 78 genes in the YAP/TAZ Up geneset that were tested in the GSEA in Figure 7D, 56 were in the leading edge (i.e. enriched) of both the YAP/TAZ-dependent lines and the TEAD-dependent lines. An additional 10 genes were in the leading edge of either the YAP/TAZ-dependent lines or the TEAD-dependent lines. The genes in the leading edge of each plot are now listed in Table S6, Tab 4. All of these 56 genes were found to have TEAD peaks in the analysis in Figure 4D.

R2 Comment #9: Why was tophat-cufflink used instead of salmon/kallisto/deseq2 or STAR/rsem/Deseq2? The author may want to consider those in the future, along with limma-voom etc.

Response: We agree with the reviewer that salmon/kallisto/deseq2 and STAR/rsem/Deseq2 are more current pipelines for RNA-seq data analysis, and we would use these in the future. However, the RNA-seq in this study was performed several years ago and analyzed by the MIT bioinformatics core facility. At that time, tophat-cufflink was a standard pipeline used. Since much of the initial work for this paper was done based on that tophat-cufflink analysis we did not repeat the analysis using the newer pipelines. We believe that the tophat-cufflink analysis was adequate to help identify YAP-dependent genes and that our subsequent analyses with numerous other datasets validated these targets.

Please include the markers in the blue/X-ray film gel regarding the original western. 

Response: We have added protein marker labels to all of the original western blot images.  

Reviewer 3 Report

Comments and Suggestions for Authors

This manuscript by Kanai et al generates a gene signature for YAP/TAZ/TEAD pathway activity in melanoma using cell line models. They then compare their gene set to other previously published works identifying a YAP/TAZ gene signature, TEAD target genes by ChIP, and dependency using the DepMap database. This is an exceptionally well-written manuscript with solid methodology and rigorous approaches, although its novelty is somewhat low, given that YAP/TAZ/TEAD gene signatures have been investigated by other groups. A few comments are strongly suggested to improve the manuscript.

1.     The text in the figures is nearly impossible to read. The resolution is poor and text is very small and fuzzy in both the figures and the figure legends. This made it difficult to assess the presented data, particularly for heatmaps and tables. I magnified the pdf file 2x to try to read the figures, which helped some, but the printed version was nearly unreadable.

2.     In Figure 1, the authors are establishing their cell line model, with YAP mutants, to identify their YAP/TEAD gene signature. This is a nice experimental setup, but the functional data in subpanel C, showing lung metastasis incidence in a tail vein injection model, needs to be supported by images of either whole tissue or histological staining of lung tissue to visually confirm the graphed data.

3.     The overarching theme of the manuscript is that a YAP/TAZ/TEAD gene signature could be used to predict cancer sensitivity to TEAD small molecule inhibitors as a treatment strategy. While the authors use DepMap data to support this assertion, ideally it would be confirmed by performing viability assays with a TEAD inhibitor in cell lines that are positive and negative for the authors’ gene signature.

Reviewer 4 Report

Comments and Suggestions for Authors

- The authors identified a set of genes regulated by TEADs in cancer cells and found that the levels of these genes could predict if the cancer cells require TEADs for survival and growth.

- The article is novel and interesting and clearly fills a gap in the current literature.

- Introduction is well written.

- Methods are adequate.

- Results are very relevant.

- figures and tables are very informative.

- Discussion shows excellent critical points.

- Overall considered, the paper can be accpeted as it is.

- The authors should be congratulated for their work.

Author Response

We thank the reviewer for their time and support of our manuscript. 

Round 2

Reviewer 3 Report

Comments and Suggestions for Authors

The authors have appropriately addressed this reviewer's concerns.